# Abiotic Community Constraints in Extreme Environments: Epikarst Copepods as a Model System

**Tanja Pipan** [1,*]**, Mary C. Christman** [2] **and David C. Culver** [3]

1   ZRC SAZU Karst Research Institute, Novi trg 2, SI-1000 Ljubljana, Slovenia
2   Departments of Biology and Statistics, University of Florida, MCC Statistical Consulting LLC, Gainesville, FL 32611, USA; marycchristman@gmail.com
3   Department of Environmental Science, American University, 4400 Massachusetts Ave. NW, Washington, DC 20016, USA; dculver@american.edu
*   Correspondence: tanja.pipan@zrc-sazu.si; Tel.: +386-5-700-1940

**Abstract:** The general hypothesis that the overall presence or absence of one or more species in an extreme habitat is determined by physico-chemical factors was investigated using epikarst copepod communities as a model system, an example of an extreme environment with specialized, often rare species. The relationship between the presence or absence of epikarst copepods from drips in six Slovenian caves and 12 physico-chemical factors (temperature, conductivity, pH, $Ca^{2+}$, $Na^+$, $K^+$, $Mg^{2+}$, $NH_4^+$, and $Cl^-$, $NO_2^-$, $NO_3^-$, and $SO_4^{2-}$) was explored. Statistical analyses included principal components analysis, logistic mixed models, stepwise logistic multivariate regression, classification trees, and random forests. Parametric statistical analyses demonstrated the overall importance of two variables—temperature and conductivity. The more flexible statistical approaches, namely categorical trees and random forests, indicate that temperature and concentrations of $Ca^{2+}$ and $Mg^{2+}$ were important. This may be because they are essential nutrients or, at least in the case of $Ca^{2+}$, its importance in molting. The correlation of $Cl^-$ and $NO_3^-$ with copepod abundance may be due to unmeasured variables that vary at the scale of individual cave, but in any case, the values have an anthropogenic component. This contrasts with factors important in individual species' niche separation, which overlap with the community parameters only for $NO_3^-$.

**Keywords:** extreme environment; niche; epikarst; classification trees; Copepoda; random forests

## 1. Introduction

### 1.1. Background

Extreme environments, such as polar ice caps, hypersaline lakes, acidic peatlands, hyper-arid deserts, groundwater aquifers, and caves, often harbor a specialized fauna that is both numerically rare and patchily distributed. Rarity is typical in extreme environments for which resource scarcity is at least part of what makes them extreme. This resource scarcity is, with the exception of chemoautotrophic sites [1] and sites very near the surface [2], a universal characteristic of aphotic habitats, which lacks primary productivity. In addition, sites with extremes of temperature, such as polar ice caps, or aridity, such as hyper-arid deserts, are also typically very low primary productivity habitats. Sampling rare species and populations of such extreme environments presents unique statistical challenges itself [3], and analysis of ecological relationships among species and between species and their environment is especially difficult given the low numbers of individuals of any one species. Whilst classic ecological techniques like Canonical Correspondence Analysis [4] and Outlying Mean Index [5] address the relationship between species occurrences and abiotic parameters, the spatial distribution of taxa

(i.e., occupancy) is still an unexplored question. This question is interesting because it addresses the question of the overall constraints to adaptation to extreme environments.

In extreme environments, there is often a suite of species that show specialized physiological, morphological, and behavioral adaptations to the environment. There are numerous examples in other extreme environments, such as stomatal modification of desert dwelling plants [6], and heat-resistant enzymes in species living in thermal vents on the ocean floor [7]. Convergent adaptation in extreme environments is perhaps best developed, or at least best studied, in the numerous species found in caves and other subterranean habitats. Species specialized for and isolated in subterranean environments typically have elaborated extra-optic sensory structures as well as reduced or absent eyes and pigment [8]. This convergent morphology is an expression of the constraints imposed by aphotic environments. Such communities of specialists, whether in thermal vents, caves, or other extreme environments, are often cited as examples of convergence and parallelism [9–13]. The study of the whole community of morphologically convergent species, rather than individual species, has yielded insights into not only adaptation but also community function (e.g., [14,15]).

### 1.2. Epikarst as a Model Extreme Environment

Epikarst is the shallow part of karst areas, where stress release, climate, tree roots, and geological processes (especially dissolution of rock) fracture and enlarge rock joints and cracks, creating a more porous zone than the underlying limestone [16]. It is directly beneath the soil, and retains water from infiltration, typically in a 3–10 m thick zone (Figure 1). It is subject to periodic drying and to periodic flushing of water that dislodge the plankton sized organisms that inhabit it [17]. Epikarst is aphotic, low in organic carbon [18] with habitat spaces that limit size and shape of animals occupying the space [19].

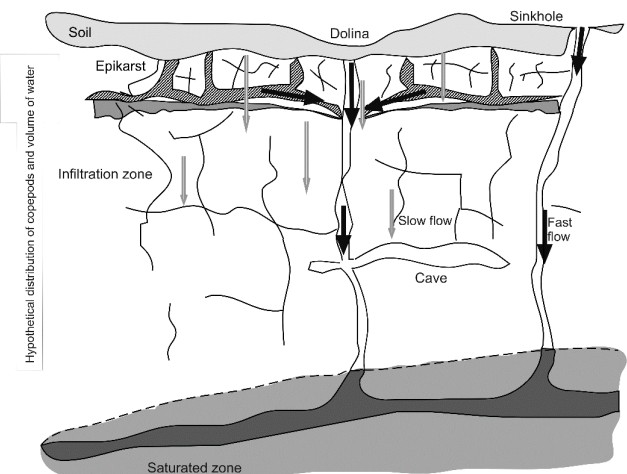

**Figure 1.** Conceptual model of epikarst. Gray arrows indicate the direction of slow water flow and black arrows are faster flow paths. From [18]. Used with permission of ZRC SAZU, Založba ZRC.

Copepods are the predominant group found in epikarst throughout the world, but other crustacean taxa occur regularly, including Amphipoda, Ostracoda, and Syncarida [2]. Sampling of the epikarst fauna is necessarily indirect, and the least biased samples are those that capture the invertebrates found in epikarst water dripping into caves [20,21]. Due to this indirect sampling, most environmental information is about the physical and chemical characteristics of the dripping water. A universal feature of such epikarst samples is that a significant number of samples have no animals (e.g., [20,22–24]).

### 1.3. Aims and Goals

The question we wish to examine is whether these rare, specialized communities are constrained in their spatial distribution by physical and chemical factors. It is common that not all extreme

environments of a particular kind, e.g., caves, harbor a specialized fauna. In any region, a number of caves have no specialized fauna, although caves without species are not always reported (but see [25]). The absence of a specialized fauna from a particular site may be because of barriers to dispersal to the site, historical accident, or because the physico-chemical conditions at the site are unsuitable for the fauna in general.

Our hypothesis is that differences in chemical and physical factors among extreme habitats provide an explanation for the presence or absence of a specialized fauna from a particular site. To test this, we analyzed an extensive data set on the occurrence of copepods specialized for life in a shallow subterranean habitat—epikarst, sampled in central Slovenia [20,26]. Pipan [26] took > 250 monthly chemical and biological samples from dripping epikarst water in six caves in central Slovenia. Slightly more than half of these samples lacked copepods. The question then is how the physico-chemical parameters differed between the two kinds of samples (those with and without fauna), and whether there was a biological explanation for the difference. These epikarst copepod communities are especially appropriate for analysis because they are diverse and with variable densities among and within subterranean patches [27]. Thus, we can not only elucidate the important community-wide parameters, but we can also compare these parameters with the individual species parameters that differentiate their niches.

## 2. Methods and Materials

### 2.1. Study Area and Sampling Procedures

The study area is in the western Dinaric Mountains of Slovenia. Six caves were studied—Črna jama, Dimnice, Pivka jama, Postojnska jama, Škocjanske jame, and Županova jama (see [20]).

Direct sampling of epikarst habitats is not possible due to the small size of the cavities and their inaccessibility. Thus, the epikarst water and epikarst fauna must be sampled indirectly by taking samples of the percolation water that drips directly from the ceiling. To do this, water from trickles was directed through a funnel into a collecting container. On two sides, the plastic containers have $5 \times 5$ cm openings covered with a 60 μm diameter mesh to retain animals in the container. At monthly intervals, the water and the organisms in the containers were collected and preserved in 4% formalin. This sample is likely biased in favor or smaller organisms since they are more easily dislodged by water currents [1,2], but it is less biased than pool samples because there is differential survival and reproduction in pools [2]. For the purpose of this study, only the presence or absence of adult copepods was used for analysis. Juveniles were not included because adults should be a better representation of a permanent population [1,2]. In addition, a series of chemical and physical measurements were made at the time of the biological sampling.

Temperature and conductivity were measured in situ by a conductivity meter (WTW[TM] Model LF 91); pH was measured by a pH meter (WTW[TM] Model 323). Water samples for ionic analysis were collected and stored in plastic containers and kept at 4 °C prior to measurement. An ion chromatograph (Metrohm[TM] Model 761 Compact IC) was used to analyze the concentrations of major ions. Chloride, nitrite, nitrate, phosphate, and sulphate were determined using an anion separation column. Sodium, ammonium, potassium, calcium, and magnesium were analyzed using a cation separation column. Concentrations of $PO_4^{3-}$ were always below the level of detectability of 0.05 mg/L, and therefore not included in the analysis. Detection limits of ions ranged from 50 to 100 μg/L.

Sampling (Table 1) began in Postojnska jama, but when five months of sampling produced relatively few copepods, sampling was switched to other parts of the Postojna-Planina Cave System (Črna jama and Pivka jama). Sampling in three other caves in the region (Dimnice, Škocjanske jame, and Županova jama) was performed for the whole period. Five drips were sampled in each cave, except in Postojnska jama, where 10 drips were sampled. The copepod fauna in the studied drips shows no temporal correlation for each drip [21], and samples within and among drips are treated as independent.

**Table 1.** Summary of sampling coverage of the six study caves, by month. An X indicates that all of the drips in the cave were sampled for physico-chemical parameters, at least once. Data from [26].

| Month | Postojnska Jama | Pivka Jama | Črna Jama | Škocjanske Jame | Dimnice | Županova Jama |
|-------|-----------------|------------|-----------|-----------------|---------|---------------|
| Jan   |                 | X          | X         | X               | X       | X             |
| Feb   |                 | X          | X         | X               |         |               |
| Mar   |                 |            |           |                 | X       | X             |
| Apr   | X               |            |           |                 |         |               |
| May   | X               |            |           | X               | X       | X             |
| Jun   | X               |            |           | X               | X       | X             |
| Jul   | X               |            |           | X               | X       | X             |
| Aug   | X               |            |           |                 |         |               |
| Sep   | X               |            |           | X               | X       | X             |
| Oct   | X               | X          | X         | X               | X       | X             |
| Nov   |                 | X          | X         | X               | X       | X             |
| Dec   |                 | X          | X         | X               | X       | X             |

## 2.2. Statistical Methods

Summary statistics and exploratory analyses.—Stratified means and standard errors were calculated for each physico-chemical variable where strata were identified as each distinct combination of cave and month. In order to explore the relationships among variables, principal components for each cave were obtained for the same set of variables, in this case ignoring the sampling site effect. Thus, we assumed that the relationships among variables are the same regardless of the cave within which the samples were collected. A mixed model with the interaction of month and cave as a fixed effect and sampling site (drip) within cave as a random effect was run for each physico-chemical variable to assess temporal pattern within each cave. The residuals from each model of the physico-chemical variables were checked to verify that the assumptions of the mixed model were met. In some cases, transformations were required to obtain normality and homogeneous variance while in some of the other variables, the model was modified to allow unequal residual variances among months. Where relevant, pairwise differences of means were tested using the Tukey–Kramer method for controlling the experiment-wise error rate.

Modeling the probability of copepod presence in a sample. We ran an all possible subsets logistic regression analysis in which every possible subset of the physico-chemical variables was used as the predictor set. The models were ordered by number of variables and value of the $\chi^2$ statistic for the overall test of significance. From these the variables identified in the most parsimonious model with the highest $\chi^2$ value, they were chosen for a logistic mixed model for predicting probability of copepod presence with month fixed and sampling site (drip) within cave as a random effect was run.

One concern is that the actual relationship between the explanatory variables and the probability of copepod presence may be different than that assumed for a logistic regression model. In the logistic regression model all variables are assumed to be additive in their effect on the probability of presence, and that the relationship can be described by a logistic curve. In reality, a variable may have a threshold effect such that there is a discontinuous jump in probability at some level of the explanatory variable, or several variables could interact in their effect on the probability. To address these issues, the non-parametric random forest and classification tree approach was used [28–30]. The recursive partitioning approach known as random forests was used to provide individual measures of variable importance [29]. This utilizes a bootstrap method that allows for the detection of the relative importance of co-varying physico-chemical measurements. The basic approach is that sets of five randomly selected variables (out of our total of 12) are used to determine a classification tree. This is repeated many times (in our case 2000). For each tree, the important variables are recorded. This allows variables that may be masked by correlation with other predictor variables to be identified as important

predictors. The best predictors from random forests were then used to construct classification trees. The predictor space is recursively partitioned into a set of rectangular areas such that samples with (or without) copepods are grouped. A probability of presence of copepods is predicted within each partition. The process is repeated until no improvement in prediction is possible. Confusion matrices showing the accuracy of prediction of the models were calculated. All computations were done in either R (cran.r-project.org), SAS© v.9.2 (SAS Institute, Cary, NC, USA), or JMP®v9.0.0 (SAS Institute, Cary, NC, USA).

## 3. Results

### 3.1. What Are the Physical and Chemical Conditions Present in Epikarst Water?

Table 2 summarizes the basic statistical properties of the stratified samples of physico-chemical variables. Overall, the physico-chemical values are typical for carbonate and calcium rich and slightly alkaline waters found in karst regions (Table 2). As is typical of carbonate waters, calcium dominated the cations, and pH was slightly above neutrality. Conductivity was high, a reflection of the relatively high concentrations of $Ca^{2+}$. The median temperature (8.9 °C) was very similar to the mean annual temperature for the region, approximately 9.0 °C.

**Table 2.** Basic statistics for measured variables, using stratified estimates (caves and months define the strata). There were 35 strata and 252 observations. All concentrations are in mg/L. Conductivity is in µS/cm. Temperature is in °C.

| Statistic | Temperature | Conductivity | pH | $Ca^{2+}$ | $K^+$ | $Mg^{2+}$ | $Na^+$ | $NH_4^+$ | $Cl^-$ | $NO_2^-$ | $NO_3^-$ | $SO_4^{2-}$ |
|---|---|---|---|---|---|---|---|---|---|---|---|---|
| Mean | 8.16 | 356.64 | 7.79 | 37.71 | 0.47 | 0.93 | 1.36 | 0.096 | 2.34 | 0.0030 | 2.84 | 6.18 |
| S.E. of Mean | 0.106 | 4.641 | 0.012 | 0.747 | 0.013 | 0.025 | 0.086 | 0.005 | 0.219 | 0.0004 | 0.360 | 0.135 |
| Median | 8.91 | 365.00 | 7.73 | 40.60 | 0.41 | 0.91 | 0.95 | 0.084 | 1.44 | 0.000 | 0.78 | 5.95 |
| S.E. of Median | 0.043 | 4.91 | 0.012 | 1.47 | 0.013 | 0.027 | 0.043 | 0.002 | 0.05 | 0.001 | 0.083 | 0.15 |

Table 3 shows the correlations among the 12 variables. There are four correlations greater than 0.5: $K^+$ and $NO_3^-$; $Na^+$ and $NO_3^-$; $Mg^{2+}$ and $SO_4^{2-}$; and $Na^+$ and $Cl^-$. The major cations, $Ca^{2+}$, $K^+$, $Mg^{2+}$, and $Na^+$, are positively correlated, most of them significantly so (Table 3). Overall, 19 of 66 correlations had absolute values greater than 0.25 and 32 of 66 were statistically significant.

**Table 3.** Pairwise correlations among the variables, using data from all caves and all dates. Values in bold are greater than 0.5 and values marked with an asterisk are statistically significant ($p < 0.05$).

| | $NO_3^-$ | $NO_2^-$ | $NH_4^+$ | $SO_4^{2-}$ | $K^+$ | $Ca_2^+$ | $Na^+$ | $Mg^+$ | $Cl^-$ | °C | pH |
|---|---|---|---|---|---|---|---|---|---|---|---|
| $NO_3^-$ | 1 | | | | | | | | | | |
| $NO_2^-$ | 0.202 * | 1 | | | | | | | | | |
| $NH_4^+$ | 0.140 * | 0.065 | 1 | | | | | | | | |
| $SO_4^{2-}$ | 0.075 | 0.025 | 0.061 | 1 | | | | | | | |
| $K^+$ | **0.500** * | 0.316 * | 0.327 * | 0.261 * | 1 | | | | | | |
| $Ca_2^+$ | 0.194 * | −0.034 | 0.034 | 0.492 * | 0.282 * | 1 | | | | | |
| $Na^+$ | 0.622 * | 0.198 * | 0.264 * | 0.183 * | 0.470 * | 0.210 * | 1 | | | | |
| $Mg^{2+}$ | 0.247 * | 0.023 | 0.261 * | **0.588** * | 0.420 * | 0.461 * | 0.406 * | 1 | | | |
| $Cl^-$ | 0.480 * | 0.030 | 0.162 * | 0.023 | 0.088 | 0.157 * | **0.687** * | 0.216 * | 1 | | |
| Temperature | −0.113 | 0.094 | −0.029 | 0.072 | −0.062 | −0.271 * | −0.208 * | 0.022 | −0.258 * | 1 | |
| pH | 0.121 | −0.009 | −0.035 | −0.074 | 0.156 * | −0.023 | 0.058 | −0.105 | 0.021 | −0.039 | 1 |
| Conductivity | 0.294 * | 0.055 | −0.023 | −0.228 * | −0.039 | 0.121 | 0.005 | −0.053 | 0.085 | −0.047 | −0.123 |

To investigate the differences among caves and dates for environmental variables, PCA analysis, by cave, is shown in Figure 2. Three sites (Pivka jama, Postojnska jama, and Škocjanske jame) have

no distinct clusters of points but with outliers (Pivka jama with one and Škocjanske jame with two). The other three caves (Črna jama, Dimnice and Županova jama) have two or three clusters of points, but except for Črna jama, there was no clustering of PCA scores by date. In Črna jama values for October and November formed a distinct cluster, along the first principal component. Loadings on the first component of Črna jama were highest for $Mg^{2+}$, $Ca^{2+}$, $K^+$, $NH_4^+$, $Cl^-$, and $SO_4^{2-}$. There was variation in factor loadings among caves. In general, cation and anion loadings were positive for component 1.

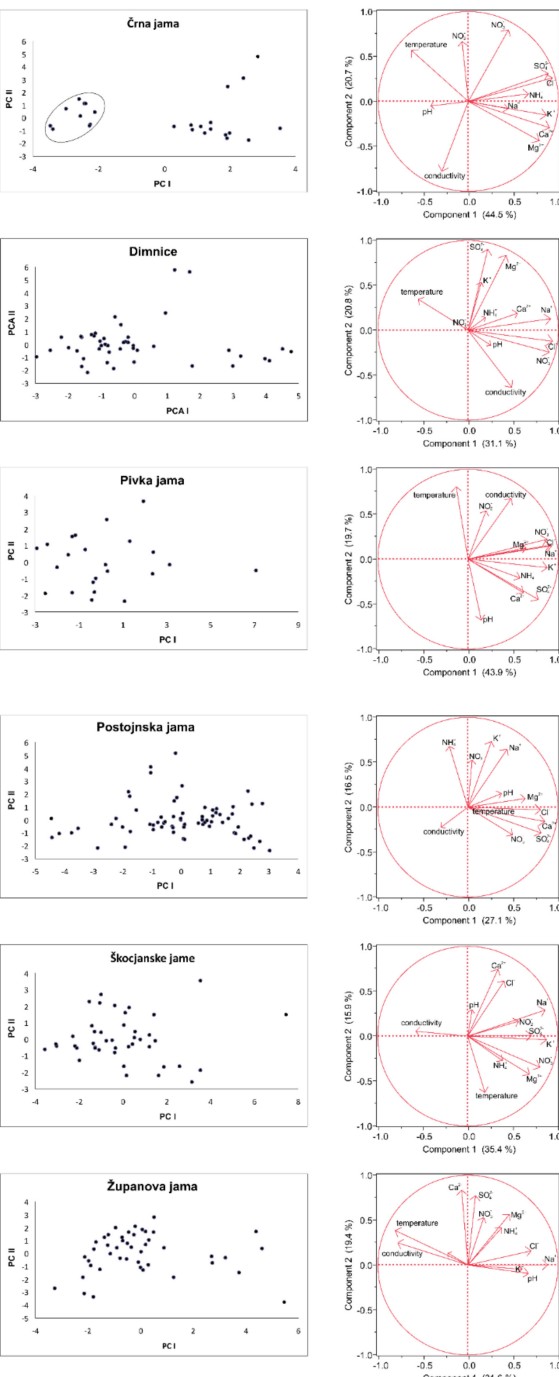

**Figure 2.** PCA analysis by cave. On the left hand panels, individual samples are shown. For Črna jama, the ellipse encloses all samples taken during October and November. It was the only such cluster of sampling dates for the six caves. See text for details. On the right hand panels the eigenvectors for the physical and chemical variables are shown.

Loadings on the second component were more variable. As Postojnska jama was sampled at different months (April through October) than Črna jama and Pivka jama (October through February, Table 1), it is interesting to compare these three PCA plots. In Postojnska jama, temperature has a very different loading than in Črna jama and Pivka jama, but this pattern is not repeated for other variables. In Postojnska jama, the first two PCA axes accounted lowest percent of the variance (43.6%) and in Pivka jama, the first two axes account for the highest percent of the variance (65.2%).

Of the 252 samples taken at various times in the six caves, 136 (54.0%) had adult copepods. For none of the six caves were there significant monthly differences in the presence or absence of copepods (Table 4). Therefore, the connection between physico-chemical factors and the presence or absence of copepods can be considered without the confounding effects of cave and time interactions.

**Table 4.** General linear mixed model type III tests of fixed effect of month/cave combination on presence/absence of copepods and on the chemical and physical variables. $NO_2^-$ was not included because it was not possible to normalize the variable. The list of caves with monthly differences includes all caves with at least one significant difference among months. Some of the results are graphically displayed in Figure 3.

| Variable | Transformation | Covariance Parameter Estimates (SE) for the Random Effects | | Numerator df | Denominator df | *F*-Value | *p* | Caves with Monthly Differences |
|---|---|---|---|---|---|---|---|---|
| | | Sampling Site within Cave | Residual | | | | | |
| Presence of copepods | Binary | 2.050 (1.193) | na | 37 | 179 | 0.74 | 0.86 | None |
| Temperature | None | 1.603 (0.462) | 1.070 (0.113) | 43 | 178.1 | 11.64 | <0.0001 | All caves except Postojnska jama and Županova jama |
| Conductivity | None | 3595.6 (1026.4) | 1935.9 (204.8) | 43 | 177.8 | 5.22 | <0.0001 | All caves except Črna jama |
| pH | None | 0.00566 (0.00239) | 0.0263 (0.0028) | 43 | 179.4 | 3.31 | <0.0001 | Pivka jama, Postojnska jama, Županova jama |
| $Ca^{2+}$ | None | 17.90 (7.37) | * | 43 | 96.5 | 16.58 | <0.0001 | All caves |
| $K^+$ | None | 0.00923 (0.00408) | ** | 43 | 93.5 | 3.00 | <0.0001 | Pivka jama, Postojnska jama, Županova jama |
| $Mg^{2+}$ | None | 0.0838 (0.0257) | 0.0772 (0.0082) | 43 | 176.2 | 5.26 | <0.0001 | All caves except Županova jama |
| $Na^+$ | Log | 0.232 (0.075) | 0.320 (0.035) | 43 | 176.4 | 4.71 | <0.0001 | All caves except Županova jama |
| $NH_4^+$ | None | 0 | *** | 43 | 105.8 | 2.09 | 0.0012 | Postojnska jama |
| $Cl^-$ | Square root | 0.258 (0.071) | 0.0841 (0.0089) | 43 | 177.3 | 4.66 | <0.0001 | All caves except Postojnska jama and Županova jama |
| $NO_3^-$ | Log | 0.888 (0.242) | 0.232 (0.025) | 43 | 177.1 | 7.77 | <0.0001 | All caves |
| $SO_4^{2-}$ | None | 1.449 (0.501) | 3.126 (0.301) | 43 | 177.2 | 9.25 | <0.0001 | All caves |

\* Unequal variances among months, with residuals ranging from 8.95 to 338.01; ** Unequal variances among months, with residuals ranging from 0.0064 to 0.0776; *** Unequal variances among months, with residuals ranging from 0.000245 to 0.00370.

In contrast, for at least one cave, average monthly values for every physico-chemical parameter showed at least one significant monthly difference (Table 4, Figure 3). There were differences among caves, and the most prominent of these anomalous patterns was Županova jama. It was unique in showing no temporal differences in either $Mg^{2+}$ or $Na^+$, and shared with Postojnska jama the lack of temporal differences in temperature and $Cl^-$ (Table 4). Postojnska jama was unique in showing temporal differences in $NH_4^+$. With the exception of temperature (Figure 3, upper panel), none of the

variables showed a yearly cyclical pattern. From April to November, temperature remained near the yearly high for each cave, and then, in all except Županova jama, fell several degrees. In the case of Postojnska jama, there were no data during the winter months. More typical was the pattern shown by conductivity (lower panel of Figure 3), with occasional erratic changes of highs and lows, except for Črna jama which had no significant monthly differences. There was no overall seasonality with respect to the appearance of highs and lows, nor was there any consistent spacing of highs and lows.

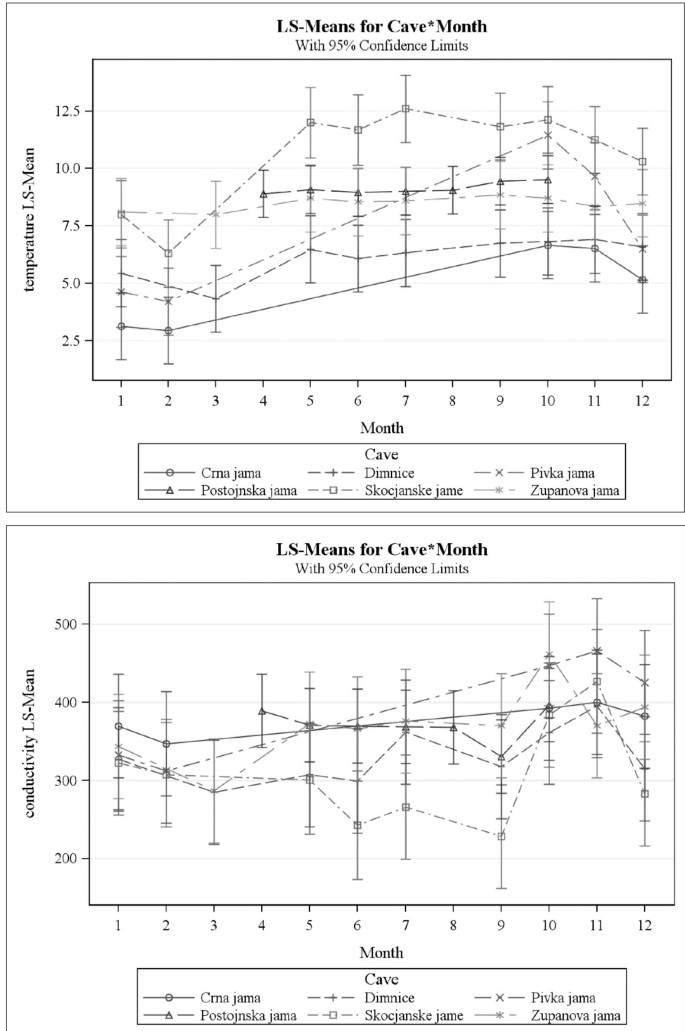

**Figure 3.** Temporal pattern of temperature (upper panel) and conductivity (lower panel). Vertical bars are 95% confidence intervals. For temperature, Črna jama, Dimnice, Pivka jama, and Škocjanske jame had significant temporal variation. For conductivity, Dimnice, Pivka jama, Postojnska jama, Škocjanske jame, and Županova jama had significant temporal variation.

## 3.2. Which Variables Could Be Important in Limiting Copepod Distribution?

The results of the all possible subsets regression analysis are shown in Table 5. No model with more than six variables gave a significant improvement in fit over the six variable model which included temperature, conductivity, $K^+$, $NO_2^-$, $Na^+$, and $Cl^-$ (Table 5). When a seventh variable is added ($Ca^{2+}$), the corresponding increase in $\chi^2$ is less than one. When the random effect of sampling site within a cave was included in the model with these explanatory variables, only conductivity and temperature show a significant effect, both being significant at $p = 0.022$.

**Table 5.** Best subset of physico-chemical data for predicting presence of copepods for each of k = 1, . . . 7 explanatory variables using logistic regression. The six variable model was chosen for further analysis because addition of an additional variable added less than one to the $\chi^2$ value.

| Number of Variables | $\chi^2$ | Variables |
|:---:|:---:|:---:|
| 1 | 12.978 | Temperature |
| 2 | 18.265 | Temperature, Conductivity |
| 3 | 22.426 | Temperature, Conductivity, $K^+$ |
| 4 | 24.350 | Temperature, Conductivity, $K^+$, $NO_2^-$ |
| 5 | 26.117 | Temperature, Conductivity, $K^+$, $NO_2^-$, $Na^+$ |
| 6 | 27.782 | Temperature, Conductivity, $K^+$, $NO_2^-$, $Na^+$, $Cl^-$ |
| 7 | 28.441 | Temperature, Conductivity, $K^+$, $NO_2^-$, $Na^+$, $Cl^-$, $Ca^{2+}$ |

A more general and realistic approach, which assumes neither independence or monotonicity of predictor variables is a random forest, showing the factors affecting the presence or absence of epikarst copepods is a random forest (Figure 4). As expected, temperature and conductivity are the most important, as was the case for the logistic regression. However, four other variables seem likely to play a role in determining copepod distribution, listed in decreasing order of importance: $Cl^-$, $NO_3^-$, $Ca^{2+}$, and $Mg^{2+}$. How these factors influence presence or absence of epikarst copepods can be determined by a classification tree using these six factors. Three of the six variables are shared with all possible subsets regression (Table 5)—temperature, conductivity, and $Cl^-$.

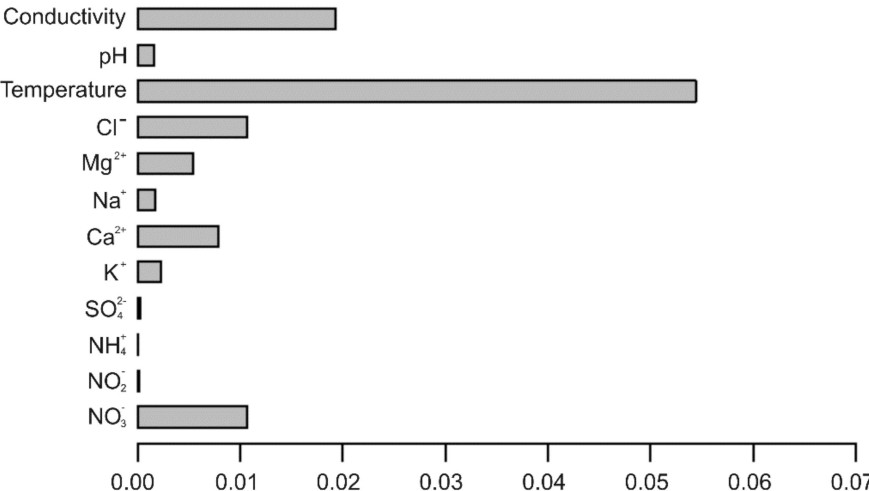

**Figure 4.** Bar graph of the relative importance of different variables in generating classification trees, when five variables are repeatedly drawn at random.

The best model from the classification tree (Table 6) is shown in Figure 5 and summarized in Table 6. Overall, the tree classified 192 of 252 samples (76%) correctly with regard to the presence or absence of copepods, and had a generalized $R^2$ of 0.45. The presence of copepods was correctly predicted 69% (94 of 136) of the time and the absence of copepods was correctly predicted 84% (98 of 116) of the time. The first branch was for temperature and 57 of 136 occurrences of copepods could be predicted solely on the basis of temperature being less than 8.2 °C (Table 6). There was no further branching on the left but for temperatures greater than 8.2 °C, $Ca^{2+}$ concentration was the next partition with high concentrations also being associated with the presence of copepods. The remainder of the partitions (Figure 5) involved $Cl^-$ (twice), $NO_3^-$, $Mg^{2+}$, and $Ca^{2+}$ (again). Of the eight terminal leafs, three were associated with absence, and the most important of these (81 of 116 samples without copepods) were samples with high temperature, intermediate $Ca^{2+}$, low $Cl^-$, low $Mg^{2+}$, and high $NO_3^-$ (Table 6).

**Table 6.** Terminal "leafs" of classification tree (see Figure 5). Conductivity differences were not in the optimal classification tree. Blanks indicate that no dichotomy for that variable exists for that leaf. Concentrations are in mg/L and temperature is in °C.

| Temperature | Ca$^{2+}$ | Cl$^-$ | NO$_3^-$ | Mg$^{2+}$ | Frequency with Copepods | Frequency without Copepods | *n* |
|---|---|---|---|---|---|---|---|
| <8.2 | | | | | 0.668 | 0.332 | 85 |
| ≥8.2 | ≥57.5 | | | | 0.758 | 0.242 | 22 |
| ≥8.2 | <15.3 | ≥2.11 | | | 0.843 | 0.157 | 9 |
| ≥8.2 | <15.3 | <2.11 | | | 0.410 | 0.590 | 22 |
| ≥8.2 | <57.5 and ≥15.3 | | <0.21 | | 0.587 | 0.413 | 15 |
| ≥8.2 | <57.5 and ≥15.3 | | ≥0.21 | ≥1.25 | 0.447 | 0.553 | 11 |
| ≥8.2 | <57.5 and ≥15.3 | ≥2.85 | ≥0.21 | <1.25 | 0.556 | 0.444 | 5 |
| ≥8.2 | <57.5 and ≥15.3 | <2.85 | ≥0.21 | <1.25 | 0.052 | 0.948 | 83 |

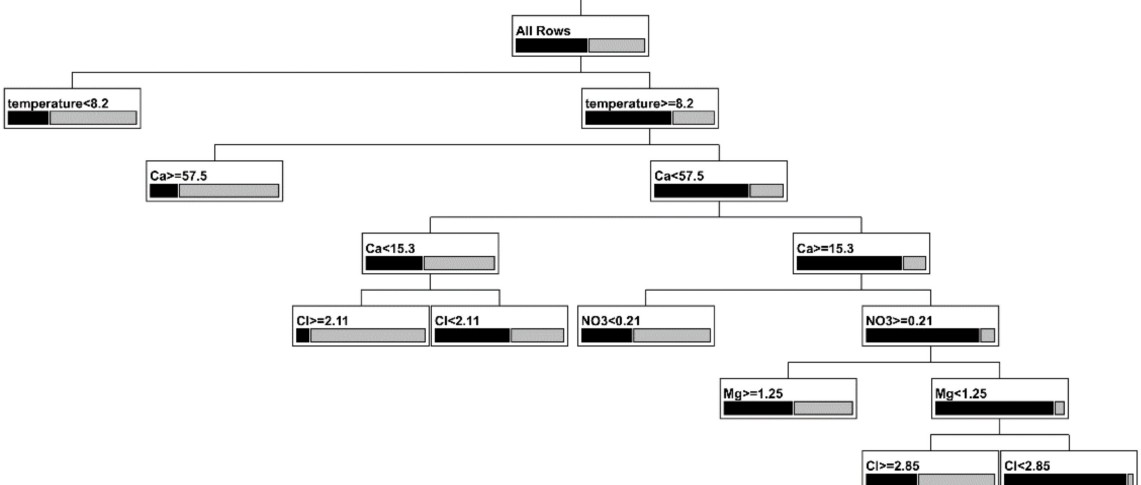

**Figure 5.** Classification tree for presence or absence of epikarst copepods. The size of the black rectangle is the relative number of empty samples and the size of the gray rectangle is the relative number of samples with copepods. See Table 6 for details.

## 4. Discussion

### 4.1. Patterns of Variation of Physico-Chemical Factors

The overall physico-chemical characteristics of epikarst drip water are that of water in contact with carbonate rock, with resulting high levels of conductivity, a predominance of Ca$^{2+}$ cations, slightly basic pH, and temperatures close to the mean annual air temperature (Table 2) [31].

The differences among samples were caused by a combination of location (Figure 2), seasonality (Table 4), and water retention time. We did not measure residence time of water in this study, but previous studies in Postojnska jama [32], found that residence time of water varied from 2.5 months to over a year. Similarly Kluge et al. [33] found retention times of one to three years in three German caves. Three caves—Dimnice, Postojnska jama, and Škocjanske jame—have relatively thick ceilings [20,26], so that, all things being equal, residence time of water may be greater in these caves. If residence time is correlated with presence (and abundance) of copepods, then differences in residence time will result in differences among the caves. The underlying geology of the caves is nearly identical, with all caves formed in Jurassic limestones, although there may be some dolomite in Županova jama [34].

There are factors that vary on the scale of the projected area of a cave onto the surface that influence the physico-chemical variables that we measured. For example, sites in Pivka jama are high in $NO_3^-$ [20]. We suspect, but cannot demonstrate, that this is the result of the presence of a campground and associated structures [35]. Other potentially anthropogenically causes include high $Cl^-$ in Dimnice [20], which is also correlated with $NO_3^-$ (Table 3), and may indicate water from wells with high concentrations of both [36]. There is no commercial activity and minimal agricultural activity in the area, but elevated levels of $Cl^-$ nonetheless suggest anthropogenic inputs because of the relative rarity of naturally occurring $Cl^-$. High $Cl^-$ concentrations may be from animal waste, human wastewater, or other sources such as salt licks.

Given variation in residence time of water underground, it is not surprising that there is almost no seasonality, but high temporal variability with respect to physico-chemical variables. One manifestation of the variability of residence time is the response to the flow rate of drips to rainfall events, according to Kogovšek [32]. After a dry period or drought, the flow rate of drips does not increase after a rainfall event, but once the epikarst layer is saturated with water, the increase of flow can be in a matter of hours. Of course, the water exiting the drip is not the precipitation water itself, but some of the water previously stored. That is, rainfall has a piston effect on the water in epikarst [37,38].

Temperature showed a rather different pattern than the chemical variables (Figure 3). The differences in temperature among caves are likely the result of details of the differences in sampling times (Table 1), climate differences (especially with respect to Škocjanske jame, which is in region of more Mediterranean climate than the others), and vertical distance from the surface. Except for Postojnska jama, for which there were no winter samples (Table 1), all caves showed a seasonal pattern of reduced temperatures from January to April (Figure 3, upper panel). In some cases, this difference was small. In Županova jama, the monthly least squares means (see Figure 3), varied less than 1 °C, but the coldest months were between January and March. While these differences were not statistically significant, the seasonal pattern in Črna jama, Dimnice, Pivka jama, and Škocjanske jame was statistically significant, and the temperature differences were greater. Kogovšek [32] also found a seasonal pattern in temperature for two drips in Postojnska jama, although the range was less than 1 °C. The relationship between temperature and residence time is a complex one (see [39–41]) for a more detailed discussion, but the temperature of resident water eventually approaches the mean annual temperature [39]. Temperature also reflects season, as seen in this study. The lower temperatures in winter are indicative of other events, such as changes in evapo-transpiration and precipitation.

*4.2. The Epikarst Copepod Physico-Chemical Niche*

A useful beginning point of the analysis is to see which individual physico-chemical factors can account for presence or absence of epikarst copepods in a sample. If the effect of cave is included as a covariate, only temperature and conductivity were significant predictors of the presence or absence of copepods in a sample. However, when a non-parametric multivariate approach, using variables shown to be important in a random forest analysis (Figure 4) in a classification tree (Figure 5, Table 6), was employed, the most important correlates of copepod presence were:

- Temperature,
- $Ca^{2+}$,
- $Mg^{2+}$,
- $Cl^-$,
- $NO_3^-$.

Conductivity itself was unimportant in the classification tree, presumably because some its major components, especially $Ca^{2+}$ and $Mg^{2+}$, were exposed by the random forest analysis, which teased apart correlated variables (see Table 3). Correlation analysis indicated that conductivity was not pairwise correlated with any of the cations (Table 3). The results of the classification tree argue that the relationship of conductivity with the other variables is more complicated than simple pairwise

relationships. In fact, a regression tree (not shown) using conductivity as the response variable and the other five variables as predictors indicated that conductivity is explained in order by $Ca^{2+}$, $Cl^-$, $NO_3^-$, and $Mg^{2+}$ which is the same order that these variables enter the tree for predicting presence of copepods.

Copepods tend to be in samples with lower temperatures (Table 6), and so the connection between temperature and copepod presence is likely a seasonal one. It is very likely that the relationship between occurrence probability and temperature is driven by copepods getting washed out of epikarst in greater numbers in winter because flow rates are greater. We did not measure flow rates but Kogovšek [32] continuously monitored discharge from two epikarst drips in Postojnska jama for a period of more than two years, and demonstrated that discharge rates were highest in the winter. Rouch [42] observed a similar pattern of copepod drift from a karst spring.

Of the other variables shown to be important in the classification tree, $Ca^{2+}$ has a strong connection with the biology of copepods. It is critical in the molting process, and some subterranean crustacean species, like the amphipod *Gammarus minus,* are limited to carbonate springs [43]. Additionally, of interest is that copepods tend not to be found in water with temperature greater than 8.2 °C and $Ca^{2+}$ concentrations greater than 57.5 mg/L. Water in epikarst can be supersaturated with respect to $Ca^{2+}$ (part of the mechanism of deposition of $CaCO_3$ in caves (e.g., stalactites)) and this may cause physiological problems for animals in this water. While carbonate geochemists have long focused on the $Ca^{2+}$ -$HCO_3^-$ system, we suggest it deserves more attention from biologists working in the same systems. $Mg^{2+}$ is also a critical nutrient [44], and it is possible that it is limiting in some contexts.

While $Mg^{2+}$ concentration may or may not be biologically significant, it seems likely that the correlation of epikarst copepod abundance and $Cl^-$ is due to some other unmeasured variable, one that varies at the scale of cave. $Cl^-$ concentrations in Dimnice are twice as high (5.54 ± 1.12 mg/L) as in any other cave [26]. We suspect that it is not $Cl^-$ that is important but some other unmeasured factor.

The correlation with $NO_3^-$ is perhaps also the result of some other unmeasured variable, but nitrate is also biologically important. It is a frequent contaminant of karst aquifers, resulting from agricultural runoff, septic tanks, and perhaps atmospheric deposition [45]. There are few studies of the nitrogen cycle in caves or epikarst, but available evidence suggests that it is not a limiting nutrient [15,46]. However, there are still a number of puzzling aspects of the nitrogen cycle (see, [45]), such as whether nitrogen fixation occurs in caves. If not, Barton [47] points out it is likely to be limiting in some circumstances.

If we take classification trees as the most general approach to the understanding of the connections of copepod occurrence to physico-chemical parameters, then we have the following factors, listed in order of importance:

1. Temperature, which is likely a reflection of flow velocities rather than community structure.
2. Calcium and perhaps magnesium ions, which are important, both as essential nutrients and in molting.
3. Anthropogenically augmented ions—$Cl^-$ and perhaps $NO_3^-$—may indicate contamination from upgradient well water, or they may be surrogates for particular epikarst sites, where some unmeasured variable is important.

The multi-faceted statistical approach, combined with an emphasis on the overall community rather than individual species, has made some sense of the complex patterns of variation of physico-chemical variables.

### 4.3. The Relationship between Community Physico-Chemical Niche and Individual Physico-Chemical Niches

Pipan [20,26] and Pipan et al. [27] analyzed the same data from a different perspective, one that emphasized niche separation among the epikarst copepod species. Using the same variables with the addition of ceiling thickness, they found that the following parameters were significant factors in distinguishing individual species in a Canonical Correspondence Analysis (CCA): ceiling thickness;

$NO_3^-$; $K^+$; and $Na^+$. Of these, only $NO_3^-$ was important on a community-wide basis. Thus, the parameters by which the species are separated are, for the most part, distinct from the parameters that predict the presence or absence of one or more species (see Figure 5).

An example of individual niche analysis for $NO_3^-$ is shown in Figure 6. This is reflected in the presence of some species, especially *Brycocamptus dacicus, Bryocamptus* n.sp., *Moraria varica,* and *Maraenobiotus brucei*, only in high $NO_3^-$ concentrations and only in Pivka jama (Figure 6). Whatever the source, it points to nitrate as an important factor in organizing communities (see [44]). What is also apparent in Figure 6 is the difficulty in separating or even characterizing the physico-chemical niches of the different species. Of the 27 species found in drips, there were data on more than 100 individuals for only three species, and only an additional four species had more than 10 individuals for which nitrate data were available.

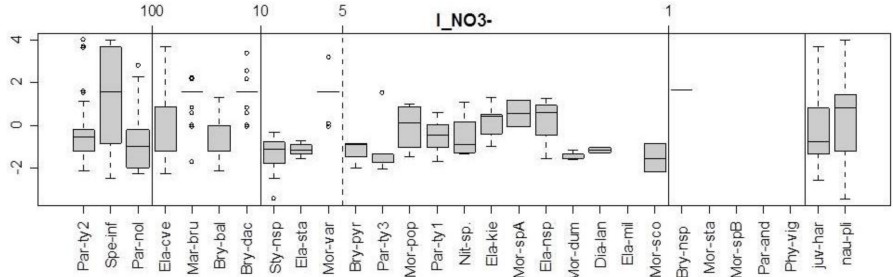

**Figure 6.** Box and whiskers plot of the log $NO_3^-$ concentrations (mg/L) for different copepod species, which are arranged by their abundance (indicated by the vertical lines). The rectangles enclose the middle 50% of the data, the line across each rectangle is the group median, the "whiskers" are the minimum and maximum values. Species names are Par-ty2: *Parastenocaris* sp.2; Spe-inf: *Speocyclopes infernus*; Par-nol: *Parastenocaris nolli alpine*; Ela-cve: *Elaphoidella cvetkae*; Mar-bru: *Maraenobiotus brucei*; Bry-bal: *Brycamptus balcanicus*; Bry-dac: *Brycamptus dacicus*; Sty-nsp: *Stygepactophanes* n.sp.; Ela-sta: *Elaphoidella stammeri*; Mor-var: *Moraria varica*; Bry-pyr: *Bryocamptus pyrenaicus*; Par-ty3: *Parastenocaris* sp. 3; Mar-pop: *Moraria poppei*; Par-ty1: *Parastenocaris* sp. 1; Nit-sp: *Nitocrella* n.sp.; Ela-kie: *Elaphoidella kieferi*; Mor-spA: *Moraria* sp. B; Ela-nsp: *Elaphoidella* n.sp.; Mor-dum: *Morarioipsis dumonti*; Dia-lan: *Diacyclops languidoides*; Ela-mil: *Elaphoidella millennii*; Mor-sco: *Morariopsis scontenophila*; Bry-nsp: *Bryocamptus* n.sp.; Mor-sta: *Moraria stankovitchi*; Mor-spB: *Moraria* sp. B; Par-and: *Parastenocaris andreji*; Phy-vig: *Phyllognathopus viguieri*.

The data we used were not originally collected for the purpose of elucidating physico-chemical niches, and this is the case for many ecological and bioinventory studies. In many of these studies data collected on basic water chemistry remains unconnected and often unanalyzed with respect to the organisms being studied. The results of our study suggest that a careful field study exploring the impact of variation in the physical and chemical characteristics of water on the likelihood of copepods being present may yield additional insights into the forces that control aquatic community structure and dynamics. These impacts could be implicit such as when variables act as surrogates for other factors (such as temperature) or explicit, i.e., with direct effects (such as is likely the case with $Ca^{2+}$). There are other constraints on the epikarst copepod community, especially the lack of light and low levels of organic carbon, that are a formidable barrier for not just copepods, but any species to survive in epikarst. Thus, the physico-chemical constraints suggested by the classification tree (Figure 5), are not absolute, but constraints in the context of no light and little organic carbon, among the extreme conditions that characterize epikarst in general.

## 5. Conclusions

A focus on general community occupancy of extreme habitats may ultimately yield insights into environmental constraints on even groups that seem ubiquitous and unconstrained by environmental factors, like copepods. There were physico-chemical differences between samples with fauna and

without fauna. The physico-chemical factors that constrained epikarst copepod communities in this study were, with the exception of $NO_3^-$, different than those that separated the physico-chemical conditions in which different species were found. Some of the community constraints were likely surrogates for age of the epikarst water; and some, like $NO_3^-$ and $Ca^{2+}$, may be important biological requirements themselves. In general, the study of such community constraints is made difficult by co-variation of explanatory parameters, and the technique of random forests and classification trees is a useful way out of this dilemma. The study of community constraints in extreme habitats is a fruitful field for further inquiry.

**Author Contributions:** Conceptualization, T.P. and D.C.C.; methodology, T.P. and D.C.C.; software and formal analysis, M.C.C.; writing—original draft preparation, review and editing, T.P., D.C.C. and M.C.C. All authors have read and agreed to the published version of the manuscript.

**Funding:** This research was funded by a Mellon Fund grant from the College of Arts and Sciences of American University for D.C.C. and T.P. was funded by EU H2020 project eLTER, RI-SI-LifeWatch and Slovenian Research Agency, P6–0119.

**Acknowledgments:** Work on this project by DCC was supported by a Mellon Fund grant from the College of Arts and Sciences of American University. Work on this project by TP was supported by the Karst Research Programme P6–0119, RI-SI-LifeWatch, and EU H2020 project eLTER.

**Conflicts of Interest:** The authors declare no conflict of interest.

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
