# Peer review of "Abiotic Community Constraints in Extreme Environments: Epikarst Copepods as a Model System"

_diversity, doi:10.3390/d12070269_

Round 1
Reviewer 1 Report
Dear Authors,
The scientific weight of this paper is exceptional for an area that is extremely little researched. Congratulations to all authors on an extremely interesting and new segment of research related to the epikarst. With extremely small corrections, the work will be excellent. I, therefore, support this extremely interesting research and recommend it for publication in Diversity journal.
All arguments and suggestions are in the manuscript (pdf)
All detailed corrections and arguments are in the enclosed manuscript (pdf).
All the best,
Reviewer

Author Response
Reviewer 1 response
Note: comments from reviewer are not sequential so some numbers are missing, but all comments answered.
General comments: The reviewer highlighted chemical formulas to ask about whether they were correct with respect to subscript and superscript. We think we did them all correctly.
Page 1, #1 A subtitle (“Background”) was added.
Page 1, #2 “Acidic peatlands” was added to list of extreme environments
Page 2, #2 Paragraphs were moved as suggested.
Page 2, #6 We do not want to remove these paragraphs. They tie in epikarst to extreme environments and provide a useful segue to the main body of the paper.
Page 3, #3,6,7,10,13 All these minor edits were done.
Page 4, #1 This sentence was removed per reviewer #3.
Page 4, #5,#8 Extra spaces were removed
Page 5, #1 Degree symbol was fixed.
Page 5, #7 We are not sure what the reviewer is asking for in the table caption. Units are all in mg/L as indicated.
Page 5, #9 All subscripts and superscripts were checked and we think they are ok.
Page 6, #1 Table was corrected and replaced.
Page 6, #4 This is covered in the discussion.
Page 6, #6 Subscripts and superscripts were checked.
Page 6, #12 Spaced removed.
Page 7, #1 Figure enlarged to fill entire page.
Page 9, #1 Unfortunately, we cannot add diacritical marks to the cave names.
Page 11, #12. Graphs are enlarged.
Page 11, #13 The spaces and lack thereof have no significance but we do not think they add confusion. It would be difficult for us to alter this.
Page 12, #2 We have added a sentence connecting ceiling height, residence time, and likely occurrence of copepods.
Page 12, #7 We were unaware of this possibility and have modified the text and added references. Reference number of following papers were changed.
Page 12, #9 A space was added.
Page 13, #8 Plus minus sign is corrected.
Sticky note—more Cl and NO3 discussion. We have not done so because this is not our area of expertise but have tried to indicate enough so that the topic could be followed up in a new study.
Page 13, #21 We have rewritten this point in order to bring it into accord with the previous argument.
Page 14, #1 We modified the sentences to reflect the previous discussion of nitrate.
Page 14, #3 We have added species names to the caption rather than make a very short appendix.
Page 14, #6 Previous arguments about nitrate are not relevant here.

Reviewer 2 Report
The paper is an important contribution to understanding the factors shaping the underground communities, in this case the rarely studied epikarst.
The paper is clearly written and readable. The balance between background, data presentation and interpretive discussion is appropriate, as well as the extension of the manuscript. The figures and tables are informative.
My suggestion is to accept it for publication, after including some information. My comments are listed below.
L71-73 You say that "The question then is how the physico-chemical parameters differed between the two kinds of samples (those with and without fauna), and whether there was a biological explanation for the difference". Can you please answer to this question more directly in the Conclusion?
L101-103 Can you please comment on how good a sample from percolating water is in representing the actual Copepod (or any other) community of epikarst? Are there any studies on that? Can you comment does it depend on the physical properties in each specific case eg. size if pores, speed and volume of water in relation to pore size, more places for fauna to hide and avoid washing out? Can it also depend on species? Ones that are bigger/have modified morphology can resist washing out better then the others so are less often sampled with this method although they might be abundant?
L107 Why only adults? Do juveniles/subadults have different biology/behavior? Did you do the analyses with juveniles included? Does it change the patterns?
L200 Were there samples with no adults but with juveniles/subadults? How many? Again, why not include them?
L276-281 Maybe I missed it, but do higher levels of NO2-, NO3- and Cl- mean lower numbers of Copepods?
L 326-331 Could this in fact mean that the correlation with temperature is just a consequence of them being more effectively washed out in the colder moths when water flow is stronger? And did you measure that, or have any data on that - is the dripping stronger in winter? Also, you don't have any biological explanation for relation with temperature?
L336 Can you add the value, what is the supersaturated Ca2+ concentration?
Figures:
Figure 1. bar on the left, showing distribution of copepods, is poorly visible
Figure 3. maybe add colors to the graph to make it easier to interpret
Author Response
Reviewer #2
- We changed to conclusion of directly answer the questions. We added a sentence (second one), and modified the third to last sentence.
- There is no direct answer to this question since epikarst cannot be sampled directly. We do add a sentence indicating that the sampling we used is better than direct sampling of pools, but that it is likely biased toward smaller organisms which are more easily dislodged.
- Indeed, one could analyze the samples by including juveniles, but we chose not to because we thought that adults were a better indication of a permanent population. An examination of the original data suggests their inclusion would change things very little, since there were only 4 samples with juveniles and no adults. We have added a sentence indicating our justification.
- See our response to #3.
- The context here is differences on the scale of caves. Their impact on copepod presence is best understood in the context of Figure 5, that is, it depends on context. NO2 is not an important determinant so we have deleted it from the sentence.
- We thought our paragraph was clear on this point. We did add a phrase to the last sentence in the paragraph indicating that we did not measure flow rates directly.
- Saturation (and supersaturation) depends on temperature and we do not think it is useful add a table or figure for this point, which detracts from the main focus of the paper.
- The high quality figure is clear.
- The graph is in color!

Reviewer 3 Report
The manuscript of Pipan et al. is very interesting and brings a novel approach on untangling the relationship between presence or absence of a diverse group (the copepods) in epikarst samples and environmental factors. I believe that the current manuscript is very well written and brings into discussion an important scientific topic.
My comments are mainly related to writing style. Please find them below:
Lines 34-35’which lacks primary producers’
Lines 39-40: ‘given the low abundance of individuals’
Lines 40-43 ‘Whist classic taxonomic techniques like multivariate ordination (e.g. CCA) or the Outlying Mean Index address the relationship between species occurrences and abiotic parameters, the spatial distribution of taxa (i.e. occupancy) is still an unexplored question’
Line 59’spatial distribution’ and ‘It is common that’
Lines 68-69’ Using a sampling 68 method described below’ can be deleted
Lines 73-75’ These epikarst copepod communities are especially appropriate for analysis 73 because Pipan et al. [19] previously analyzed the chemical differences among the individual species 74 using Canonical Correspondence Analysis (CCA).’ It could be deleted or re-explained. The epikarst copepods may be suitable because they are diverse and with variable densities among and within subterranean patches, not because the CCA relates species abundance with variation of abiotic factors
Lines 90-91’ but other 90 crustacean taxa occur regularly, including’ replaced by ‘along with…’
Line 104:’2’ replaced by ‘two’
Line 118: delete ‘The sampling scheme for the 6 caves is complicated (Table 1).’ Amd refert to table 1 in the next phrase.
Lines 168-169: delete the sentence’ Table 2 summarizes the basic statistical properties of the stratified samples of physico-chemical 168 variables’ and add the table 2 in the end of the following sentence
Lines 169-170: replace Overall, the values are typical of karst water that has been in contact with carbonate rock. 169 As is typical of carbonate waters, calcium dominated the cations, and pH was slightly above 170 neutrality ‘ with ‘Overall, the physic-chemical parameters were typical for carbonate calcium rich and slightly alkaline waters found in karst regions (Table 2)’
Lines 175-178: Only report the significant correlations and don’t forget to mention that Spearman coefficients were >0.5
Lines:181-195: The whole paragraph can be deleted, away too descriptive and not providing any meaningful information. Start with what the overall trend of the PCA shows from all caves.
Start with paragraph from Line 188
Lines 200-202: The paragraph’ If the 200 probability of the presence or absence of adult copepods was related to cave and month, then it would 201 be difficult to infer that the differences in physico-chemical parameters in samples with and without 202 copepods, are due to chemicals and not cave or time effects.’ Should be deleted. This was a question already mention in Introduction. Moreover, this is Results section.
Line 227: replace ‘a’ with’ no’
Line 266: Replace with ’differences….were caused by a combination of location, seasonality and water retention time..’
Lines 277-278: replace the sentence ‘We suspect, but cannot demonstrate, that this is the result of the presence of a 277 camp ground and associated structures’ with ‘in NO2- and NO3- [18], potentially related to increased local anthropic impact[35]’
Line 285’ according to Kogovšek [32]’
Line 302: any sound reference to back up the statement ‘but the temperature of resident water eventually approaches the mean 302 annual temperature.’ ?
Lines 312-316: I would replace the bullet points with a simple enumeration of the physic-chemical factors
Lines 322-325: ‘Actually, the conductivity is explained in decreasing importance (as reflected by Spearmen coefficients ? was it significant? Can’t see it since it was not presented) by Ca2+, Cl-, NO3-, and Mg2+, the very order revealed by random forest trees, enhancing the importance of these standalone variables in explaining copepods’ distribution ‘
Line 327: I agree that copepods dynamic is influences by seasonality, hence , replace ‘It is unlikely, but possible’ with ‘ It is very likely that the relationship between the occurrence probability of copepods and temperature is driven by’. Moreover, how about the old studies of Rouch in France, who addressed more or less the same issues, as more recent (if available) literature ?
Lines 333-334: replace ‘and some crustacean species’ with ‘stygophiles and stygobites’; otherwise, Ca is important in moulting process of most crustaceans and even molluscs.
Line 340 and 341: Please provide references and explain a bit why Mg is a critical nutrient. It it isn’t, then, please provide a reference stating that the connection between the two is not clear. Could it be a confounding effect of just water chemistry? (Ca with Mg ?)
Line 369-372: see my above comment with bullet points
Line 397: I would delete’ the lack of light’, since many taxa adapt very well to this habitat and rather conditioned by other prerequisite. However, I agree with low carbon content
Author Response
Reviewer 3 response
- Suggested change in wording was made.
- Changed to “given the low numbers of individuals”.
- Suggested change in wording was made.
- Suggested change in wording was made.
- Suggested deletion was made.
- Sentence was deleted and reviewer’s substitute sentence added.
- We prefer the original wording.
- Suggested change was made.
- Suggested change was made.
10 and 11. We incorporated all the suggested changes.
- We are not clear what the reviewer means about Spearman correlations, which were not reported. Only significant Pearson correlations were discussed.
- We disagree that this paragraph does not provide meaningful information. It summarizes the important question of site (cave) and seasonality on the results. It may be descriptive but it is important.
- We started a new paragraph.
- The sentence (but not the paragraph) was deleted.
- We did not make the change. It results in a double negative.
- The suggested changes were made.
- We prefer the original wording.
- Suggested change in wording was made.
- A reference was added. It was a previous one [38] so no change in reference numbering is required.
- We prefer to retain the bullet points to highlight their importance.
- We are not entirely clear about this comment. However, the variables (beginning with Ca2+) are not standalone variables. That is the very point of the classification tree. They are considered in combination, a point shown clearly in Table 6. So we did not make the suggested change.
- We made the suggested change and added Rouch as reference 41. The references following are renumbered.
- We added the word subterranean rather than stygophile etc. in order to avoid more jargon.
- We added a reference on magnesium (#45) and renumbered references following.
- Bullets were eliminated.
- We emphatically disagree the light is unimportant. Not to belabor the point but we have written several articles to this effect. So, we have left the phrase in.
